# The Protein Composition of Bovine Milk from Once-a-Day and Twice-a-Day Milking Production Systems in New Zealand

Marit van der Zeijden [1,2,*], Ashling Ellis [1,3,†], Nicolas Lopez-Villalobos [1,4], Siqi Li [1,†], Nicole C. Roy [1,5,6] and Warren McNabb [1,6,*]

1 Riddet Institute, Massey University, Palmerston North 4422, New Zealand
2 School of Food and Advanced Technology, Massey University, Palmerston North 4422, New Zealand
3 Smart Foods & Bioproducts Group, AgResearch Ltd., Te Ohu Rangahau Kai, Massey University, Palmerston North 4422, New Zealand
4 School of Agriculture and Environment, Massey University, Palmerston North 4422, New Zealand
5 Department of Human Nutrition, University of Otago, Dunedin 9054, New Zealand
6 High-Value Nutrition National Science Challenge, Liggins Institute, University of Auckland, Auckland 1023, New Zealand
* Correspondence: m.vanderzeijden@massey.ac.nz (M.v.d.Z.); w.mcnabb@massey.ac.nz (W.M.)
† Current address: Fonterra Research and Development Centre, Private Bag 11 029, Palmerston North 4472, New Zealand.

**Abstract:** An increasing number of dairy farmers in New Zealand (NZ) have adopted a once-a-day (OAD) milking production system, and little is known about the impact of this production system on milk protein composition. The objective of this study was to evaluate the effect of OAD milking on the protein composition in milk from individual cows. Milk was sampled in early, mid-, and late lactation from cows kept at Massey University farms Dairy No. 1 (OAD milking) and Dairy No. 4 (TAD milking) in Palmerston North, NZ. The yields of total milk and milk solids, the proximate composition, and the protein composition were determined. Results showed that OAD milking yielded less milk and milk solids than TAD milking. However, no significant differences in protein, fat, and lactose contents were found. While the proportions of total casein (CN), total whey proteins, αs1-CN, β-CN, and β-lactoglobulin were not affected by the milking frequency, milk from a OAD milking system contained higher proportions of αs2-CN and κ-CN and lower proportions of α-lactalbumin. These proteins also changed differently throughout the milking season in a OAD milking system than in a TAD milking system. These changes in the protein composition of the milk observed in a OAD milking system could have implications for its processing properties and product quality.

**Keywords:** milking frequency; once-a-day milking; seasonal variation; protein composition

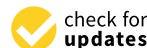



## 1. Introduction

Most farms in New Zealand (NZ) follow a twice-a-day (TAD) milking production system. A once-a-day (OAD) milking production system is increasing in popularity because of a lack of good quality long-term staff and a change in farmer lifestyle expectations, among other reasons [1–3]. In the 2018–2019 milking season, 8% of dairy farms in NZ milked their cows using a OAD milking system [4]. The total milk yield, as well as protein and fat yields, decreased when the milking frequency was reduced [5–7]. Milking frequency is an important consideration for the farmer since the value of milk to the farmer is based on the yield of milk solids, which includes fat and protein. Previous research has also shown an increase in protein and fat contents in milk from cows milked OAD compared to milk from cows milked TAD [6,8]. Another study found a change in expression levels of genes encoding for the major milk proteins [9]. This finding suggests a difference in the protein composition of the milk, but this has not been confirmed or measured in previous studies.

In addition to the milking frequency, other factors also impact milk composition. One of these factors is the stage of lactation. In NZ, most farms work with a seasonal calving

system where calving happens from late winter to spring. As a result, the meteorological seasons are aligned with the milking season and, therefore, the stage of lactation of the cow. From calving, the protein and fat contents decrease until peak lactation around 6–8 weeks postpartum before slowly increasing again. Closer to drying off, the proportion of protein and fat increases more significantly, alongside a decrease in milk yield [10–12]. The breed of the cow also affects milk composition. For example, Jersey cows produce milk that is higher in protein and fat and has a larger casein-to-whey ratio [13,14].

Milk composition, including protein composition, is an essential contributor to the processing properties of the milk. Therefore, the aim of this study was to understand the consequences of a OAD production system on the protein composition of individual cows. In this study, the milk from cows from a OAD milking system was compared to the milk from cows from a TAD milking system. Firstly, the overall impact of OAD milking on the milk and milk solids yield and the proximate composition across the milking season was investigated. Secondly, the average protein composition across the lactation was determined. Finally, the effect of the stage of lactation on milk yield and the proximate and protein composition of milk from a OAD and TAD milking system was analyzed.

## 2. Materials and Methods

### 2.1. Cow Characteristics

Cows from two Massey University research farms (Palmerston North, NZ), one utilizing a OAD (Dairy No. 1) and one utilizing a TAD (Dairy No. 4) milking production system, were selected based on their genetics, breeding worth, and expected calving date. The OAD farm switched to OAD milking in 2013, and since then, cows not suitable for OAD have been culled, and cow replacements have been produced by the superior cows for OAD milking and sires selected on a OAD selection index.

Nine cows from each system were selected, and each group had an even distribution of breeds (Holstein-Friesian (F, n = 3), Holstein-Friesian × Jersey cross (F × J, n = 3), and Jersey (J, n = 3)). The cows selected in the TAD system were used for each sampling, but some cows selected in the OAD system were excluded from the study during the season due to either mastitis or treatment for other health issues (e.g., lameness). These cows were replaced with cows under the OAD system that matched the criteria for the study.

Milk was sampled nine times throughout the milking season: three were scheduled at the beginning, three in the middle, and three towards the end. Samples were taken from the entire milking. The milk samples were categorized based on the days in milk (DIM) of the cow on the day of sampling, as early (<90 DIM), mid (90–180 DIM), and late (>180 DIM) lactation. To create a 24 h milk sample for the TAD-milked cows, the milk from the afternoon and the following morning was mixed in the laboratory proportional to the respective yields.

### 2.2. Milk Characterization

The milk yield of each cow was determined by recording the weight on the farm immediately after milking. The proximate composition of the milk was analyzed using a MilkoScan FT1 (Foss Analytics, Hillerød, Denmark). The obtained values for protein, fat, and lactose contents were used together with the total milk yield to calculate the yield of each component in kilograms.

The protein composition was analyzed using the HPLC modified from the method by Bobe et al. [15]. Samples were prepared as follows: 0.5 mL of sample was mixed with 0.5 mL of Solution A (21 mg/mL Bis-Tris, 573 mg/mL guanidine HCl, 1.57 mg/mL trisodium citrate, and 3 mg/mL dithiothreitol in MiliQ water, pH 7.0). After 1 h at room temperature, the samples were centrifuged for 5 min at $14,100 \times g$. 300 μL was taken and mixed with 900 μL of Solution B (430 mg/mL guanidine HCl in 10% acetonitrile in MiliQ water with 0.1% trifluoroacetic acid, pH 2.0). Finally, the samples were filtered through a 0.2 μm RC syringe filter before analysis. An Aeris Widepore 3.6 μm XB-C18 RP column (Phenomenex, Torrance, CA, USA) was used. The total run time was 45 min at a flow rate

of 0.6 mL/min. The separation gradient was started with solvent B (water/acetonitrile, 1:9; 0.1% trifluoroacetic acid) set at 27%, followed by an increase to 32% in 2 min, then to 45.6% in 29 min, then to 50.2% in 1 min followed by a hold at 50.2% for 2 min, and was then returned to 27% in 2 min and held for 9 min. The UV wavelength was set at 220 nm for protein detection. The peaks of each protein were identified using a standard solution of $\alpha$-CN (4 mg/mL, containing $\alpha_{s1}$-CN and $\alpha_{s2}$-CN), $\beta$-CN (3 mg/mL), $\kappa$-CN (1.5 mg/mL), $\beta$-lactoglobulin ($\beta$-LG, 1 mg/mL), and $\alpha$-lactalbumin ($\alpha$-LA, 0.5 mg/mL) (all from Sigma-Aldrich, MO, USA), and the area under the peaks were used to quantify each protein in the milk samples. The glycosylated $\kappa$-CN peaks were also identified and divided by the total peak area for $\kappa$-CN to obtain the glycosylation degree.

### 2.3. Statistical Analysis

The data was analyzed using SAS version 9.4 (SAS Institute Inc., Cary, NC, USA). A mixed model (using the MIXED procedure) was used to obtain the least square means and standard errors of the parameters analyzed in this study. The following model was used:

$$Y_{ijkl} = \mu + M_i + B_j(M_i) + S_k + M_iS_k + B_j(M_i)S_k + \beta_1 p_{ijkl} + \beta_2 p^2_{ijkl} + \beta_3 d_l + C_l + e_{ijkl} \quad (1)$$

$Y_{ijkl}$ is the observation for the trait for milking frequency i, breed j, lactation stage k, and cow l

$\mu$ is the population mean

$M_i$ is the fixed effect of milking frequency i (i = OAD and TAD)

$B_j(M_i)$ is the fixed effect of breed j nested in milking frequency i (j = F, F × J and J)

$S_k$ is the fixed effect of stage of lactation k (k = early, mid and late)

$\beta_1$ and $\beta_2$ are the regression coefficients of the linear and quadratic effects of parity p (years) of cow l

$\beta_3$ is the regression coefficient of the linear effect of deviation (days) from herd median calving date d of cow l

$C_l$ is the random effect of cow l

$e_{ijkl}$ is the residual random error assumed with mean zero and variance $\sigma^2_e$.

Marginal means and standard errors for each level of the fixed effects were obtained and used for multiple mean comparisons using Fisher's least significant difference test. Significant differences were declared at $p < 0.05$ and trends at $p > 0.05$ but lower than $p < 0.10$.

## 3. Results

### 3.1. Descriptive Statistics

The average parameters of the cows used in the study are set out in Table 1. The median calving date was taken from the entire herd on each farm. For the OAD farm, the date was 7 August 2020; for the TAD farm, the median calving date was 27 July 2020. Table 1 also shows the mean parity, breed proportions, and breeding worth at the start of the season. All cows, except three cows from the TAD milking system (second and sixth parity), were in their third or fourth parity.

Table 2 shows the descriptive statistics of the daily milk and milk solids yield, proximate milk composition, and protein composition of all samples. Table 3 shows the F-values and significance levels of the fixed effects in the statistical model used in this study for the different traits analyzed. For each trait analyzed in this study, the order of the effect of parity was included. A linear effect and a quadratic effect were tested, and the closest fit was used in the model described above. A quadratic effect was the best fit for all traits except the lactose and $\alpha$-lactalbumin contents. Therefore, only the linear effect was included in the model, and no F-value of the quadratic effect for these two characteristics is included in Table 3. A larger F-value indicates a larger contribution of a certain parameter from the model. For example, the stage of lactation generally affects all traits more than the

deviation from the median calving date (d, Table 3). And milking frequency has a greater effect on protein yield than on fat yield.

**Table 1.** Number of records and descriptive statistics of cows milked once a day (OAD) and twice a day (TAD).

| | Milking Frequency | |
| --- | --- | --- |
| | **OAD** | **TAD** |
| Number of observations | 81 | 81 |
| Deviation from the median calving date (days) | $17 \pm 13$ | $15 \pm 10$ |
| Parity (years) | $3.4 \pm 0.5$ | $3.8 \pm 1.3$ |
| Proportion of Friesian | 0.481 | 0.563 |
| Proportion of Jersey | 0.519 | 0.438 |
| Breeding worth at the start of the season | $139 \pm 32$ | $148 \pm 33$ |

**Table 2.** Descriptive statistics of daily milk yield, proximate composition, and protein composition of milk from cows once a day (OAD) and twice a day (TAD).

| Trait | N | Mean | SD | CV% | Min | Max |
| --- | --- | --- | --- | --- | --- | --- |
| Milk yield (kg/cow/day) | 162 | 22.1 | 7.4 | 33.7 | 9.3 | 42.0 |
| Protein yield (kg/cow/day) | 162 | 0.80 | 0.21 | 26.1 | 0.39 | 1.28 |
| Fat yield (kg/cow/day) | 162 | 1.04 | 0.32 | 30.3 | 0.44 | 2.22 |
| Lactose yield (kg/cow/day) | 162 | 0.99 | 0.35 | 35.2 | 0.39 | 1.95 |
| Protein content (%) | 162 | 3.84 | 0.46 | 11.9 | 2.84 | 5.02 |
| Fat content (%) | 162 | 5.02 | 0.94 | 18.7 | 2.64 | 9.00 |
| Lactose content (%) | 162 | 4.58 | 0.15 | 3.3 | 4.22 | 5.02 |
| Casein (%) | 162 | 80.0 | 3.8 | 4.7 | 73.1 | 87.7 |
| $\alpha_{s1}$-casein (%) | 162 | 22.6 | 1.6 | 7.2 | 19.1 | 27.3 |
| $\alpha_{s2}$-casein (%) | 162 | 5.7 | 1.1 | 19.1 | 3.6 | 8.9 |
| $\beta$-casein (%) | 162 | 37.4 | 2.4 | 6.5 | 31.4 | 43.8 |
| $\kappa$-casein (%) | 162 | 14.3 | 1.5 | 10.7 | 9.9 | 17.1 |
| Glycosylated $\kappa$-casein (%) | 162 | 43.6 | 5.3 | 12.1 | 29.8 | 57.3 |
| Whey protein (%) | 162 | 20.0 | 3.8 | 18.9 | 12.3 | 26.9 |
| $\beta$-lactoglobulin (%) | 162 | 16.9 | 3.9 | 23.2 | 8.7 | 23.9 |
| $\alpha$-lactalbumin (%) | 162 | 3.1 | 0.4 | 14.4 | 1.9 | 4.0 |

**Table 3.** F-values and significant levels for factors affecting the composition of milk from cows milked once a day (OAD) and twice a day (TAD).

| Trait | Factor [1] | | | | | | | |
|---|---|---|---|---|---|---|---|---|
| | **M** | **B** | **S** | **M × S** | **B × S (M)** | **p** | **p²** | **d** |
| Milk yield | 15.6 *** | 4.9 ** | 457.3 *** | 2.9 | 6.8 *** | 2.4 | 2.4 | 0.1 |
| Protein yield | 20.7 *** | 5.1 ** | 213.4 *** | 0.3 | 2.4 * | 2.1 | 2.0 | 1.4 |
| Fat yield | 6.2 * | 0.4 | 140.6 *** | 4.4 * | 2.5 * | 0.8 | 0.7 | 0.0 |
| Lactose yield | 15.6 *** | 4.5 ** | 405.2 *** | 1.7 | 5.6 *** | 2.4 | 2.4 | 0.3 |
| Protein content | 0.2 | 0.7 | 202.5 *** | 6.2 ** | 2.2 * | 0.4 | 0.4 | 0.4 |
| Fat content | 0.5 | 3.2 * | 8.3 *** | 1.5 | 0.7 | 0.1 | 0.2 | 0.0 |
| Lactose content | 0.2 | 2.6 * | 62.8 *** | 6.5 ** | 4.0 *** | 7.0 ** | | 2.3 |
| Casein | 0.2 | 1.8 | 40.7 *** | 1.0 | 0.3 | 0.1 | 0.1 | 0.3 |
| αs1-casein | 1.6 | 0.8 | 51.5 *** | 0.5 | 0.5 | 1.3 | 1.3 | 1.7 |
| αs2-casein | 15.9 *** | 2.4 | 36.0 *** | 3.7 * | 0.8 | 0.9 | 1.2 | 0.1 |
| β-casein | 1.0 | 2.0 | 28.4 *** | 2.8 | 1.2 | 1.7 | 1.7 | 0.0 |
| κ-casein | 10.7 ** | 8.7 *** | 8.3 *** | 4.0 * | 3.5 ** | 6.4 * | 6.6 * | 0.0 |
| Glycosylated κ-casein | 3.8 | 2.2 | 90.7 *** | 3.3 * | 2.5 * | 1.2 | 1.2 | 0.3 |
| Whey protein | 0.2 | 1.8 | 40.7 *** | 1.0 | 0.3 | 0.1 | 0.1 | 0.3 |
| β-lactoglobulin | 0.1 | 2.0 | 60.3 *** | 0.5 | 0.4 | 0.1 | 0.1 | 0.4 |
| α-lactalbumin | 6.4 * | 4.2 ** | 58.3 *** | 3.4 * | 0.9 | 7.7 ** | | 0.1 |

[1] M = milking frequency (OAD and TAD), B = breed (Holstein-Friesian, Friesian-Jersey crossbreed, and Jersey), S = stage of lactation (early, mid, and late), p = parity (2–6), d = deviation from median calving date of the herd for each farm. Statistical significance is given as * $p < 0.05$, ** $p < 0.01$, and *** $p < 0.001$.

### 3.2. Diet Composition

Like most farms in NZ, both Massey University dairy farms have a pasture-based feeding system. The diet composition and chemical composition of the diet offered to the cows prior to sampling are presented in Table 4 as the mean of the different rations offered on the days before the sampling day. Overall, the chemical composition of the diets fed on each farm was similar. However, the total DM did differ between the farms and was higher for the TAD farm, with 18 to 24 kg DM per cow per day, whereas the OAD farm offered approximately 16 to 17 kg DM per cow per day. In addition, the crude protein per kg DM was higher in early and mid-lactation in the OAD feed compared to the TAD feed. In contrast, neutral detergent fiber content was lower in mid- and late lactation and soluble sugar and starch contents were lower in each stage of lactation at the farm utilizing a OAD milking system.

**Table 4.** Diet composition and the chemical composition of the diet offered per cow per day at Massey University Dairy Farms No.1 (once a day) and No. 4 (twice a day).

|  | OAD [8] | | | TAD [9] | | |
|---|---|---|---|---|---|---|
|  | **Early** | **Mid** | **Late** | **Early** | **Mid** | **Late** |
| *Chemical composition* | | | | | | |
| ME [1] (MJ/kg DM) | 11.41 | 11.19 | 10.61 | 11.36 | 11.13 | 10.90 |
| CP [2] (g/100 g DM) | 22.18 | 21.40 | 19.71 | 20.86 | 19.66 | 19.57 |
| NDF [3] (g/100 g DM) | 37.55 | 40.94 | 41.61 | 37.53 | 44.12 | 44.74 |
| ADF [4] (g/100 g DM) | 19.10 | 21.38 | 25.57 | 17.28 | 21.19 | 25.68 |
| SSS [5] (g/100 g DM) | 16.00 | 11.65 | 11.38 | 18.68 | 16.27 | 13.92 |
| Lipid (g/100 g DM) | 4.44 | 4.58 | 4.34 | 3.60 | 4.22 | 4.55 |
| *Diet composition (fed prior to sampling days on DM basis, kg DM/cow/day)* | | | | | | |
| Pasture | 11.7 | 11.3 | 7.3 | 17.0 | 17.0 | 7.5 |
| Chicory | 1.3 | 2.7 | 1.0 | - | - | - |
| Maize silage | 1.0 | - | 0.3 | 4.3 | 5.0 | 2.7 |
| DDG [6] | 1.8 | - | - | - | 1.0 | 0.5 |
| Tapioca | 0.8 | - | 1.3 | - | - | - |
| Molasses | - | - | - | 1.0 | - | - |
| Concentrates [7] | - | - | 2.0 | 1.0 | - | 0.7 |
| Dry roughage | - | - | - | 0.1 | - | 0.1 |
| Baleage | - | 2.0 | 4.0 | 0.7 | 1.0 | 6.2 |

[1] ME = metabolizable energy, [2] CP = crude protein, [3] NDF = neutral detergent fiber, [4] ADF = acid detergent fiber, [5] SSS = soluble sugars and starch, [6] DDG = distillers dried grain, [7] either soy- or corn-based concentrate, [8] OAD = once a day, [9] TAD = twice a day.

### 3.3. Effect of Milking Frequency on Milk Yield, Proximate, and Protein Composition

The estimated marginal means and standard errors for the different milk characteristics from cows milked OAD and TAD are presented in Table 5. The milk yield shows the total milk volume per cow per day in liters. In the OAD milking system, cows yielded 17.0 L a day on average, compared to a significantly higher volume of 25.2 L in the TAD milking system. The total protein, fat, and lactose yields from the OAD cows were also significantly lower than those from the TAD cows, by 29%, 26%, and 33%, respectively. The mean protein content of the milk was 3.90% in OAD milk and 3.78% in TAD milk, but the milking frequency effect was not significant. The mean fat content (OAD, 5.17%, TAD, 4.83%) and lactose content (OAD, 4.56%, TAD, 4.58%) in the milk did not differ between the two milking systems.

**Table 5.** Estimated marginal means and standard errors of daily milk yield, proximate composition, and protein composition of milk from cows once a day (OAD) and twice a day (TAD).

| Trait | Milking Frequency | | *p*-Value | |
|---|---|---|---|---|
| | OAD | TAD | Milking Frequency | Stage of Lactation |
| Milk yield (L) | $17.0 \pm 1.1$ | $25.2 \pm 1.2$ | <0.001 | <0.001 |
| Protein yield (kg) | $0.65 \pm 0.03$ | $0.92 \pm 0.03$ | <0.001 | <0.001 |
| Fat yield (kg) | $0.87 \pm 0.07$ | $1.18 \pm 0.07$ | 0.014 | <0.001 |
| Lactose yield (kg) | $0.78 \pm 0.05$ | $1.16 \pm 0.06$ | <0.001 | <0.001 |
| Protein content (%) | $3.90 \pm 0.14$ | $3.78 \pm 0.15$ | 0.657 | <0.001 |
| Fat content (%) | $5.17 \pm 0.25$ | $4.83 \pm 0.26$ | 0.464 | <0.001 |
| Lactose content (%) | $4.56 \pm 0.03$ | $4.58 \pm 0.03$ | 0.693 | <0.001 |
| Casein (%) | $80.5 \pm 1.6$ | $79.2 \pm 1.8$ | 0.680 | <0.001 |
| $\alpha_{s1}$-casein (%) | $21.8 \pm 0.6$ | $23.3 \pm 0.7$ | 0.216 | <0.001 |
| $\alpha_{s2}$-casein (%) | $6.7 \pm 0.3$ | $4.7 \pm 0.3$ | <0.001 | <0.001 |
| $\beta$-casein (%) | $36.5 \pm 0.9$ | $38.2 \pm 1.0$ | 0.312 | <0.001 |
| $\kappa$-casein (%) | $15.5 \pm 0.4$ | $13.1 \pm 0.4$ | 0.001 | <0.001 |
| Glycosylated $\kappa$-casein (% of total $\kappa$-CN) | $46.8 \pm 1.6$ | $41.0 \pm 1.7$ | 0.054 | <0.001 |
| Whey protein (%) | $19.5 \pm 1.6$ | $20.8 \pm 1.8$ | 0.680 | <0.001 |
| $\beta$-lactoglobulin (%) | $16.5 \pm 1.7$ | $17.6 \pm 1.8$ | 0.729 | <0.001 |
| $\alpha$-lactalbumin (%) | $2.9 \pm 0.1$ | $3.2 \pm 0.1$ | 0.013 | <0.001 |

The average protein composition across the milking season in milk from OAD and TAD-milked cows is shown in Table 5 as a percentage of the sum of all proteins measured by HPLC. The total whey protein percentage given is the sum of $\beta$-LG and $\alpha$-LA, and the total CN percentage is the sum of $\alpha_{s1}$-CN, $\alpha_{s2}$-CN, $\beta$-CN, and $\kappa$-CN. There was no significant effect of milking frequency on the total CN and the total whey protein percentage in the milk, nor the $\alpha_{s1}$-CN, $\beta$-CN, and $\beta$-LG contents. The proportion of $\alpha_{s2}$-CN in the milk increased by 43% with a decreased milking frequency from 4.7% to 6.7% (OAD vs. TAD, $p < 0.001$). The $\kappa$-CN percentage was also higher ($p = 0.001$) in OAD milk (15.5%) than in TAD milk (13.3%) by 18%. There was a trend for a higher glycosylation degree of $\kappa$-CN in OAD milk than the TAD milk ($p = 0.054$). The difference in the proportion of $\alpha$-LA was significant ($p = 0.013$) and approximately 9% lower in OAD milk (2.9%) than in TAD milk (3.2%).

### 3.4. The Effect of Stage of Lactation on the Milk from OAD and TAD-Milked Cows

The previous section focused on the different characteristics of milk from the OAD and TAD milking production systems across different stages of lactation. In this section, the effect of the stage of lactation on milk (solids) yield, the proximate composition, and protein composition in each milking system are detailed. The stage of lactation influenced each of the traits analyzed in the study, as seen in Table 3. Table 6 shows the milk and milk solids yield per cow per day, the proximate composition, and the protein composition in early, mid-, and late lactation for OAD and TAD milk. The last column also shows the interaction effect between milking frequency and the stage of lactation.

**Table 6.** Estimated marginal means and standard errors of daily milk yield, proximate composition, and protein composition of milk from cows once a day (OAD) and twice a day (TAD) in early, mid- and late lactation.

| | OAD | | | TAD | | | M × S [1] |
|---|---|---|---|---|---|---|---|
| | Early | Mid | Late | Early | Mid | Late | Effect |
| Milk yield (L) | 21.8 ± 1.1 [bc] | 17.6 ± 1.2 [d] | 11.5 ± 1.2 [e] | 31.0 ± 1.2 [a] | 24.9 ± 1.2 [b] | 19.6 ± 1.2 [cd] | 0.061 |
| Protein yield (kg) | 0.80 ± 0.03 [bc] | 0.65 ± 0.04 [d] | 0.51 ± 0.03 [e] | 1.08 ± 0.04 [a] | 0.90 ± 0.04 [b] | 0.78 ± 0.04 [c] | 0.728 |
| Fat yield (kg) | 1.10 ± 0.07 [bc] | 0.89 ± 0.08 [de] | 0.62 ± 0.07 [f] | 1.48 ± 0.07 [a] | 1.08 ± 0.07 [bd] | 0.97 ± 0.07 [ce] | 0.015 |
| Lactose yield (kg) | 1.02 ± 0.05 [bc] | 0.79 ± 0.06 [d] | 0.51 ± 0.05 [e] | 1.45 ± 0.06 [a] | 1.14 ± 0.06 [b] | 0.90 ± 0.06 [cd] | 0.185 |
| Protein content (%) | 3.72 ± 0.14 [bcd] | 3.67 ± 0.15 [bcd] | 4.29 ± 0.15 [a] | 3.54 ± 0.16 [d] | 3.69 ± 0.16 [c] | 4.09 ± 0.16 [ab] | 0.003 |
| Fat content (%) | 5.11 ± 0.26 [ab] | 5.06 ± 0.28 [ab] | 5.34 ± 0.26 [ab] | 4.86 ± 0.27 [a] | 4.48 ± 0.27 [b] | 5.15 ± 0.27 [a] | 0.228 |
| Lactose content (%) | 4.69 ± 0.03 [a] | 4.54 ± 0.03 [b] | 4.46 ± 0.03 [c] | 4.65 ± 0.03 [a] | 4.56 ± 0.03 [b] | 4.53 ± 0.03 [bc] | 0.002 |
| Casein (%) | 81.1 ± 1.6 [abc] | 81.1 ± 1.7 [abc] | 79.3 ± 1.7 [def] | 80.1 ± 1.8 [ad] | 79.5 ± 1.8 [be] | 78.1 ± 1.8 [cf] | 0.359 |
| $\alpha_{s1}$-casein (%) | 22.7 ± 0.6 [ace] | 21.8 ± 0.7 [bdf] | 21.0 ± 0.6 [g] | 24.0 ± 0.7 [ab] | 23.4 ± 0.7 [cd] | 22.4 ± 0.7 [efg] | 0.618 |
| $\alpha_{s2}$-casein (%) | 6.2 ± 0.3 [b] | 6.9 ± 0.3 [a] | 7.1 ± 0.3 [a] | 4.5 ± 0.3 [d] | 4.6 ± 0.3 [d] | 5.1 ± 0.3 [c] | 0.027 |
| β-casein (%) | 36.6 ± 0.9 [be] | 37.3 ± 1.0 [ad] | 35.5 ± 0.9 [cf] | 38.4 ± 1.0 [abc] | 38.6 ± 1.0 [abc] | 37.7 ± 1.0 [def] | 0.066 |
| κ-casein (%) | 15.5 ± 0.4 [a] | 15.2 ± 0.4 [b] | 15.6 ± 0.4 [a] | 13.3 ± 0.4 [c] | 12.9 ± 0.4 [d] | 13.0 ± 0.4 [d] | 0.020 |
| Glycosylated κ-casein (% of total κ-CN) | 43.1 ± 1.7 [cdf] | 46.4 ± 1.8 [be] | 51.0 ± 1.7 [a] | 38.5 ± 1.8 [f] | 40.7 ± 1.8 [de] | 43.8 ± 1.8 [bc] | 0.039 |
| Whey protein (%) | 18.9 ± 1.6 [abc] | 18.9 ± 1.7 [abc] | 20.7 ± 1.7 [def] | 19.9 ± 1.8 [ad] | 20.5 ± 1.8 [be] | 21.9 ± 1.8 [cf] | 0.359 |
| β-lactoglobulin (%) | 15.7 ± 1.7 [adf] | 15.9 ± 1.7 [adf] | 18.0 ± 1.7 [bce] | 16.6 ± 1.8 [ef] | 17.3 ± 1.8 [cd] | 19.0 ± 1.8 [ab] | 0.623 |
| α-lactalbumin (%) | 3.2 ± 0.1 [a] | 2.9 ± 0.1 [b] | 2.7 ± 0.1 [c] | 3.3 ± 0.1 [a] | 3.3 ± 0.1 [a] | 2.9 ± 0.1 [b] | 0.036 |

[1] Interaction between milking frequency (M; OAD and TAD) and stage of lactation (S; early, mid, and late). [a, b, c, d, e, f, g] Means with different superscripts within one row are significantly different ($p < 0.05$).

### 3.4.1. Milk and Milk Solids Yield

Overall, the yield, both in liters and in terms of the different major components, decreased throughout the season and was higher in early lactation for both milking systems (OAD, 21.8 L, TAD, 31.0 L) and lower in late lactation (OAD, 11.5 L, TAD, 19.6 L). The total milk yield from OAD cows compared to TAD cows was 30% and 29% lower in early and mid-lactation, respectively (Table 6). The largest difference in milk yield was observed in late lactation, where the total milk yield was 41% lower in the OAD compared to the TAD milking system. The difference in protein yield between OAD and TAD milking systems at different stages of lactation was smaller than for the total milk yield. In early and mid-lactation, the protein yield was 26% and 28% lower in the OAD milking system than in the TAD milking system. In late lactation, this difference was 35%. The changes in the lactose yield were similar to the changes in the total milk volume yield. In early and mid-lactation, the lactose yield was 30% and 31% lower in the OAD milking system, respectively. This difference was 43% in late lactation. The total lactose yield decreased more between mid- and late lactation in the OAD milking system than in the TAD milking system. No significant interaction effects were found for the yields of milk volume, protein, or lactose.

As shown in Table 6, an interaction effect between milking frequency and stage of lactation for the total fat yield indicates a different stage of lactation effect for OAD and TAD systems. In the OAD milking system, the fat yield decreased more evenly between the different stages of lactation by 0.21 kg and 0.27 kg per cow per day between early and mid-lactation and between mid- and late lactation, respectively. On the contrary, in the TAD milking system, the fat yield decreased much more between early and mid-lactation with a 0.40 kg reduction, while between mid- and late lactation, the total fat yield per cow per day only decreased by 0.11 kg. Overall, the total fat yield fell more over the whole lactation in OAD milking than in TAD milking. The difference in fat yield between OAD and TAD-milked cows was 26%, 18%, and 35% in early, mid-, and late lactation, respectively.

### 3.4.2. Proximate Milk Composition

In both production systems, the protein content was the highest in late lactation (OAD, 4.29%, TAD, 4.09%), and the stage of lactation effect was significant. Milk from the TAD milking system had the lowest protein content at early lactation (3.54%), which was significantly different ($p < 0.001$) from the protein content at mid-lactation (3.69%). In OAD milk, however, the protein content in early and mid-lactation was similar, 3.72% and 3.67%, respectively. There was no significant difference between the protein content of OAD and TAD milk at each stage of lactation.

In OAD milk, the effect of the stage of lactation on the fat content was less evident than in TAD milk. The fat content in OAD milk at different stages of lactation did not differ significantly from each other while they did in TAD milk. The fat content of the TAD milk decreased significantly from early lactation to mid-lactation and then increased significantly in late lactation. The fat content was not significantly different between OAD and TAD milk at any stage of lactation. For both production systems, the highest fat content was in late lactation (OAD, 5.34%, TAD, 5.15%) and the lowest in mid-lactation (OAD, 5.06%, TAD, 4.48%).

In early lactation, the lactose content was the highest both in the OAD milking system (4.69%) and the TAD milking system (4.65%). In OAD milk, the lactose content decreased throughout the milking season, with the lowest value observed in late lactation (4.46%). In TAD milk, the lactose content in early lactation differed from that in mid- (4.56%), and late lactation (4.53%), but the lactose content at mid- and late lactation was not significantly different. There was an interaction effect between milking frequency and the stage of lactation for the lactose content. The lactose content in OAD milk decreased more throughout the lactation than in TAD milk.

### 3.4.3. Protein Composition

The estimated marginal means of the proportion of each of the major milk proteins and the total CN and whey protein contents at different stages of lactation are presented in Table 6. There was a significant stage of lactation effect on each of the individual proteins analyzed in the study. Milk in late lactation contained the lowest proportion of CN (OAD, 79.3%, TAD, 78.1%) and the highest proportion of whey proteins (OAD, 20.7%, TAD, 21.9%) across the lactation in both OAD and TAD milking systems. In OAD milk, the CN content in early and mid-lactation was similar, while in TAD milk, the CN content was higher in early than in mid-lactation. However, there was no significant interaction effect between milking frequency and the stage of lactation.

The $\alpha_{s1}$-CN content was the highest in early lactation (OAD, 22.7%, TAD, 21.0%) and the lowest in late lactation (OAD, 21.0%, TAD, 22.4%). There was no significant difference between the OAD and TAD milk at any stage of lactation. For the $\alpha_{s2}$-CN, an interaction effect between milking frequency and the stage of lactation was found. Above, the results showed a significantly higher proportion of $\alpha_{s2}$-CN in OAD milk compared to TAD milk. Although the lowest $\alpha_{s2}$-CN content was in early lactation for both milking systems, the content increased significantly in mid-lactation in OAD milk from 6.2% to 6.9% but remained similar in TAD milk with 4.5% and 4.6%, respectively. There was a significant increase in late lactation compared to mid-lactation in TAD milk (5.1%) but not in OAD milk (7.1%). The proportion of β-CN in OAD milk was significantly different in early, mid-, and late lactation, with the highest value in mid (37.3%) and the lowest value in late lactation (35.5%). The β-CN content in TAD milk was also lowest in late lactation (37.7%) but did not differ significantly between early (38.4%) and late (38.6%) lactation.

As with the $\alpha_{s2}$-CN content, there was a significant interaction effect between milking frequency and stage of lactation for both the proportion and glycosylation degree of κ-CN. In a OAD milking system, the κ-CN concentration was lowest in mid-lactation (15.2%) and was similar in early (15.5%) and late (15.6%) lactation. In comparison, in TAD milk, the proportion of κ-CN was highest in early lactation (13.3%) and similar in mid- (12.9%) and late lactation (13.0%). At each stage of lactation, the proportion of κ-CN was higher in OAD than in TAD milk. The glycosylation degree of κ-CN was most strongly affected by the stage of lactation of all individual proteins, as indicated by the higher F-value shown in Table 3. In both the OAD and TAD milking systems, the seasonal change in the κ-CN glycosylation degree was similar, with the highest values in late lactation (OAD, 51.0%, TAD, 43.8%) and the lowest in early lactation (OAD, 43.1%, TAD 38.5%). In late lactation, the glycosylation degree of κ-CN was significantly ($p = 0.021$) higher in OAD than in TAD milk.

The proportion of β-LG followed a similar pattern throughout the season as that of total whey proteins. The β-LG concentration was highest in late lactation for both milking frequencies (OAD, 18.0%, TAD, 19.0%). While the difference between early (15.7%) and mid (15.9%) lactation in OAD milk was not significant, in TAD milk, the proportion of β-LG was lower in early (16.6%) than in mid- (17.3%) lactation. There was an interaction effect between milking frequency and stage of lactation for the proportion of α-LA. In a OAD milking system, the α-LA content decreased significantly between each stage of lactation from (3.2%, 2.9%, and 2.7% in early, mid-, and late lactation, respectively). In a TAD milking system, only the α-LA content in late lactation (2.9%) was significantly lower than in early and mid-lactation (both 3.3%). Additionally, the proportion of α-LA was lower in OAD than in TAD milk in mid- and late lactation but not in early lactation.

## 4. Discussion

### 4.1. Effect of Milking Frequency on Milk Yield

The decrease in yield, which was 33% lower in the OAD than in the TAD milking systems, found in this study is consistent with the literature. Results from previous studies range from 8% [16] to 50% [17] in the yield loss that accompanies a reduced milking frequency. Most research, including some performed in NZ, shows a decrease of approximately 20–30% [6,18,19]. Edwards [19] compared OAD and TAD milking systems

four years before and four years after dairy farms switched from TAD to OAD. Edwards [19] found that in the first year of OAD milking, the milk yield decreased by 11% compared to the previous years and was 22% lower than the TAD farms. Compared to these results, the decrease in milk yield in the present study was higher than previously recorded in NZ [3]. Breed, between-year variation, and stage of lactation are likely factors contributing to the difference between studies. Some studies referenced above are full lactation studies, but some were only conducted at a certain stage of lactation.

As with the milk volume yield, the loss in yields of protein, fat, and lactose are on the higher end of the range previously reported. Since lactose is highly correlated with milk yield, the loss in lactose yield is expected to be like the total milk volume yield. With an expected increase in protein and fat contents, the decrease in the yield of these two components will not be as large as the total milk yield. In some studies, the decrease in fat yield was also lower than the decrease in protein yield [6,8,20]. In other studies, the opposite was observed [21,22]. In NZ, the payout is based on protein and fat yields. Therefore, decreasing the total protein and fat yield per cow will result in a lower pay-out for the farmer, although a OAD milking system also has lower overall costs [23].

OAD milking is sometimes implemented when feed availability is low or to improve the body condition score of the cows [1] and is often accompanied by lower supplement use and stocking rate [3]. However, there is little evidence to show that OAD-milked cows have lower DM requirements [3]. The average feed offered at each stage of lactation displayed in Table 4 is the amount offered to the cows, whereas the actual intake per animal probably differs. Therefore, there would have been variation in feed and energy intake between the animals on the same farm despite the same feeding regime.

### 4.2. The Overall Effect of OAD Milking on the Concentration of Major Components

The protein and fat contents in the milk were expected to increase in cows milked OAD. However, there was no effect of the milking frequency on either of these components. Previous studies have found an increase in the protein content ranging from 3% to 8% [8,20,21]. On the other hand, the increase in fat content showed more variation in previous research with a range of 1–10% [8,20,21]. This larger variation in the increase in fat content was also seen in the present study, where the coefficient of variation was higher for the fat content than for the protein content, as shown in Table 2. In contrast to the protein and fat contents, the lactose content was previously found to decrease when the milking frequency is reduced from TAD to OAD [16,20,21]. But this difference was also not always significant [8], which agrees with the results in the present study.

### 4.3. The Effect of OAD Milking on the Protein Composition

In the present study, the proportion of CN and whey proteins was not affected by the milk frequency. In contrast, many studies [24–26], but not all [8], reported a decrease in the CN-to-whey ratio in the milk with less frequent milking. The CN and whey protein fractions have been found to increase in previous research with decreasing milking frequency, likely caused by an increase in total protein concentration as milk volume decreases. A larger increase in the whey protein content contributed to a change in the ratio between the two groups of protein in the milk. This increase in whey protein content has been attributed to increased permeability of the mammary tissue due to the loss of tight junctions leading to an increased influx of serum proteins when the milking frequency is reduced [17].

In the present study, the $\alpha$-LA concentration in milk proteins was lower in OAD than in TAD milk. $\alpha$-LA is involved in lactose production, and lactose regulates milk volume by creating osmotic pressure between the blood and alveoli [27,28]. Therefore, the downregulation of milk production when the milking frequency is reduced could be correlated with the lower $\alpha$-LA concentration in the OAD milk. However, an increase in $\alpha$-LA after 7 and 15 days of decreased milking frequency of late lactation cows was also found [29]. Some authors reported an increase in the gene expression of $\alpha$-LA in the mammary tissue with increasing milking frequency [9,30]. Murney et al. [9] also found an

increase in gene expression of $\alpha_{s1}$-CN, $\beta$-CN, and $\beta$-LG in the udder half that was milked four times daily compared to the other half that was milked once daily. The authors did not look at $\kappa$-CN, but the increase in the other major proteins with a higher milking frequency may help explain the higher $\kappa$-CN content in OAD than in TAD milk. However, their results do not provide any information on the ratio between the different proteins, and the exact mechanism behind changes in the protein profile in milk as a result of such changes in the gene expression remains unclear.

The higher $\alpha_{s2}$-CN and $\kappa$-CN contents in the OAD milk than in the TAD milk. Although it is unclear what mechanism causes this increase, the change in the proportion of $\kappa$-CN potentially impacts heat stability and gelation properties of the milk through the formation of complexes with whey proteins upon heating [31,32]. To a lesser extent, $\alpha_{s2}$-CN is also involved in forming aggregates with whey proteins during heat treatment of milk [33]. Additionally, the glycosylation of $\kappa$-CN was higher in OAD than in TAD milk and higher than values previously reported in NZ [10]. Elevated levels of $\kappa$-CN glycosylation in late lactation were negatively correlated with favorable acid gelation properties of milk [34]. Cases et al. [35] also showed a negative impact of a higher $\kappa$-CN glycosylation degree on acid gelation.

### 4.4. The Effect of Stage of Lactation on the Milk from OAD and TAD-Milked Cows

#### 4.4.1. Milk and Milk Solids Yield

The difference in milk yield between OAD and TAD milk is within the previously reported range of 20–30% in early and mid-lactation, but in late lactation, the difference is much higher than found in other studies [6,18,19]. While the milk yield in the TAD milking system decreases by approximately 20% between each stage of lactation, in the OAD milking system, the milk yield decreases by 19% between early and mid-lactation and by 35% between mid and late lactation. Previous studies have reported larger production losses in early and mid-lactation than in late lactation with less frequent milking [3,36,37]. This observation regards the scenario where the milking frequency is reduced temporarily. The cows in the present study were milked OAD throughout the entire lactation and have been so for previous milking seasons. Few studies have reported the impact of lactation persistency in cows milked OAD compared to cows milked TAD [38,39]. Hickson et al. [38] found that TAD-milked cows tend to have better persistency than OAD-milked cows, while Lembeye et al. [39] found the opposite.

The loss in fat yield because of a decreased milking frequency was larger in late than in early and mid-lactation. This finding contrasts with previous studies that showed higher fat yield loss in early and mid-lactation compared to late lactation [36,37]. The results from the present study also suggest a higher persistency in OAD-milked cows than in TAD-milked cows. Both higher [6] and lower [38] persistency of milk fat yield in OAD compared to TAD milking was found in earlier studies. The fat content decreased significantly in TAD milk between early and mid-lactation, while this remained the same in OAD milk. Since the trend in milk yield throughout the lactation was similar in OAD and TAD milk, this change in the fat content likely drove the difference in the lactation curve for the fat yield.

#### 4.4.2. Proximate Milk Composition

In late lactation, milk from both TAD and OAD milking production systems had the highest protein and fat contents. The protein and fat contents increase throughout lactation until drying off [10,11,40]. TAD milk followed the same pattern, but OAD milk remained similar in protein content in the first few months of lactation. Li et al. [10] also found no difference between the protein content in early and mid-lactation in one of the two milking seasons studied. Previous studies have found that the lactose content decreases towards the end of lactation [10,11,40]. It has been suggested that a reduction in milking frequency leads to an enhanced involution process [39,41,42], decreasing the number of epithelial cells and thus decreasing lactose and total milk production. Whether this is the case when cows are milked OAD throughout the entire season is unclear.

### 4.4.3. Protein Composition

In line with the lactational change for the total CN content observed in the present study, previous research has also shown a decrease in the CN fraction in milk in late lactation compared to earlier stages [43]. This decrease has been attributed to the involution process, weakening the tight junctions and increasing the influx of serum protein [17,24]. However, some studies did not find a difference in the CN content at different stages of lactation [10,40,44].

Although similar changes over the lactation were previously observed [10] for the $\alpha_{s1}$-CN and $\alpha_{s2}$-CN contents in the milk, the stage of lactation effect on the κ-CN content in bovine milk is inconsistent across the literature. Li et al. [10] studied the seasonal change of the main proteins in milk from TAD-milked cows in NZ over two consecutive milking seasons. While the seasonal change in κ-CN content observed in OAD milk in the present study was in line with the results Li et al. [10] reported for the first year, the trend in TAD milk contrasts with their findings from either of the two milk seasons studied. O'Connell et al. [45] found an increase in the κ-CN content between mid- and late lactation. They also studied a herd managed in a seasonal calving system that was mostly grass fed during mid-lactation and the start of late lactation [45]. In another study, no significant effect of the stage of lactation on the κ-CN content was found [46]. Ostersen et al. [47] found a decrease in the κ-CN content throughout lactation. The increase in the glycosylation degree of κ-CN throughout lactation is consistent across the literature [10,48,49]. The mechanism behind the increasing glycosylation degree remains to be investigated, but there is evidence that genetics contribute to the variation [48].

Finally, the increase in β-LG towards the end of lactation was observed by some research but not all. Li et al. [10] found an increase in the β-LG content throughout the lactation in one of the two milking seasons studied. O'Connell et al. [45] found an increase between mid- and late lactation for β-LG variant B but not for variant A. In agreement with the present study, Li et al. [10] also reported a decrease in the α-LA content throughout lactation. The involvement of α-LA in the regulation of lactose production could explain the decrease in α-LA in late lactation with a decline in milk volume [10,28,50].

## 5. Conclusions

This study compared the protein composition of milk from a OAD milking system with that from a TAD milking system. The yield, proximate composition, and protein composition of milk produced by a OAD milking system and a TAD milking system were compared across different stages of lactation. The OAD system had lower yields of milk and milk solids than the TAD system. Proximate composition was not significantly different between the two systems. The effects of milking frequency, stage of lactation, and interaction between these two factors were significant for the proportions of κ-CN, $\alpha_{s2}$-CN, and α-LA in total protein. Milk from a once-a-day milking system had higher proportions of κ-CN and $\alpha_{s2}$-CN and lower proportions of α-LA than milk from a twice-a-day milking system. These proteins also changed differently throughout the milking season in a OAD milking system than in a TAD milking system. The findings from this study contribute to the understanding of the effect of OAD milking on protein composition. Such changes have potential implications for processing, for example for the gelation and heating properties.

**Author Contributions:** M.v.d.Z., A.E., N.L.-V., S.L., N.C.R. and W.M. contributed to the conceptualization, methodology, investigation, data curation, formal analysis, validation, project administration, and review and editing. Original draft, M.v.d.Z. Supervision, A.E., N.L.-V., S.L., N.C.R. and W.M. Funding acquisition, W.M. All authors have read and agreed to the published version of the manuscript.

**Funding:** This research was funded by the New Zealand Ministry of Business, Innovation, and Employment research program "New Zealand Milks Mean More", contract number MAUX1803. Stipend of M.Z. was supported by the Riddet Institute, a New Zealand Centre of Research Excellence (CoRE) funded by the Tertiary Education Commission (Wellington, New Zealand).

**Institutional Review Board Statement:** Ethical review and approval were waived for this study because this study did not directly deal with the animals; instead, milk was collected from the milking parlour as part of the usual herd-testing routine.

**Informed Consent Statement:** Not applicable.

**Data Availability Statement:** The data presented in this study are available on request from the corresponding author.

**Conflicts of Interest:** The authors declare no conflict of interest.

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
