# Peer review of "The Protein Composition of Bovine Milk from Once-a-Day and Twice-a-Day Milking Production Systems in New Zealand"

_2624-862X, doi:10.3390/dairy4040047_

Round 1

Reviewer 1 Report

Comments and Suggestions for Authors

Dear Editor,

Manuscript Title:  “The protein composition of bovine milk from once-a-day and 2 twice-a-day milking production systems in New Zealand”

The article includes results showing the effect of milking frequency on milk yield and quality throughout lactation.

I have some comments to the authors as follows:

1.     In the article, the purpose should be given in detail.

2.     The number of animals used in the experiment could have been more.

3.     The number of milking could have been 3 per day, so that more solid results could be obtained.

4.     Morning and evening milk could also be examined separately, why were samples taken by mixing them? Is this very true?

5.     Table headings should be shortened. There is no need to repeat words that are already known in the article. For example, "Massey University dairy farms in New Zealand" does not need to be repeated, it is already in the material section.

6.     Conclusions: It should include comments and information that are more detailed and based on research results.

7.     The references section should be checked according to the journal's spelling rules.

8.     Some corrections will be seen on the PDF article.

    Best regards,

Reviewer 2 Report

Comments and Suggestions for Authors

To the Authors,

The development of milk production is determined by economic and organizational factors, including those related to the frequency of milking. As the authors rightly noticed, dairy farms are increasingly switching to milking milk once a day.

The manuscript is well written, but I have reservations.

In L54-56 “However, as far as the authors are aware, no studies report the protein profile of milk from cows milked OAD compared to TAD, across the lactation”Is’nt the thrue, because I found a few similar articles.

Please find literature indicating the influence of cow breed on changes in the amount of casein and whey proteins and add it to the introduction.

About the content In the Milk Sampling section - you describe the cows selected for the experiment, so I suggest changing the name to Cow characteristics or more precisely.

When presenting the results, I suggest the amount of individual components in protein milk [g/l] - this information could increase the application value of the work and increase the frequency of citations in the future.

Please explain where the 81 observations came from? The experiment involved 9 cows in 3 different lactation periods, which gives 27 observations.

Last sentence in the Conclusions section The results of this study contribute both to the understanding of the mechanisms underlying the effects of milking...? please refer to the appropriate mechanisms.

Kind

Round 2

Reviewer 1 Report

Comments and Suggestions for Authors

Dear authors,

The article saw some improvements, 

but, 

1. I still have concerns about sampling. It is a fact that taking a sample by mixing the milk obtained after two milkings is not a correct method. Milk yield can already be calculated. However, the compositions of milk milked twice compared to single milking could be compared separately. Although the microbiological properties of milk have not been examined, it is known that it will change a lot with waiting. pH, acidity etc. These are parameters that change rapidly after milking. Therefore, it is not a correct method to examine two milking milks by mixing them in this research.

2. Regarding the purpose of the research; the aim is not only to learn the effect of the number of milkings on the composition. Also, what does this contribute to science and practice?

3. The materials and methods section is very confusing and difficult to follow because it is written in its entirety. The content should be given in small paragraphs or headings. Sample collection can be shown schematically.

4. Regarding the insufficient number of samples; authors answer is not a very valid explanation. Because there was time to do research after COVID-19.

5. Revisions marked on the PDF should be made.
